# Carbapenem-Resistant Gram-Negative Bacteria-Related Healthcare-Associated Ventriculitis and Meningitis: Antimicrobial Resistance of the Pathogens, Treatment, and Outcome

Yi Ye,[a] Yueyue Kong,[a] Jiawei Ma,[a] Guangzhi Shi[a]

[a]Department of Critical Care Medicine, Beijing Tiantan Hospital, Capital Medical University, Beijing, China

**ABSTRACT** Carbapenem-resistant Gram-negative bacteria (CRGNB)-related health care-associated ventriculitis and meningitis (HCAVM) is dangerous. We aimed to report the antimicrobial resistance of the pathogens, treatment, and outcome. All cases with CRGNB-related HCAVM in2012–2020 were recruited. Antimicrobial agents were classified as active, untested, or inactive using antimicrobial susceptibility tests. The treatment stage was classified as empirical or targeted according to the report of pathogens. The treatment effect was classified as ineffective or effective according to HCAVM-related parameters. Overall, 92 cases were recruited. For most antimicrobial agents, the resistance rate was higher than 70.0%. The polymyxin resistance rate was the lowest at 11.6%. The chloramphenicol, trimethoprim-sulfamethoxazole, amikacin, levofloxacin, and tetracycline resistance rates were relatively low, ranging from 21.1% to 64.1%. The meropenem resistance rate was 81.9%. There was no significant trend for any antimicrobial agent tested. Meropenem was the most common antimicrobial agent used in empirical treatment; trimethoprim-sulfamethoxazole and polymyxin were the most used active antimicrobial agents, and meropenem/sulbactam and polymyxin were the most used untested antimicrobial agents in targeted treatment. In total, 42 (45.7%) cases received ineffective treatments. The ineffective treatment rate of cases that received active antimicrobial agents was lower than that of cases that received untested antimicrobial agents and cases that received inactive antimicrobial agents (29.3% [12/41] versus 46.2% [18/39] versus 100.0% [12/12], $P < 0.001$). Antimicrobial resistance was prevalent but without increasing trends. Active antimicrobial agents are necessary. Additionally, untested antimicrobial agents, including meropenem/sulbactam and polymyxin, might be optional. Inactive antimicrobial agents must be replaced.

**IMPORTANCE** Carbapenem-resistant Gram-negative bacteria-related health care-associated ventriculitis and meningitis is a clinical threat because of the poor outcome and challenges in treatment. We reached several conclusions: (i) the antimicrobial resistance of pathogens is severe, and some antimicrobial agents represented by polymyxin are optional according to the antimicrobial susceptibility tests; (ii) in the background that the portion of carbapenems resistance in Gram-negative bacteria is increasing, there is no increasing trend for the antimicrobial resistance of carbapenem-resistant Gram-negative bacteria in the 9-year study; (iii) meropenem is the main antimicrobial agent in treatment, and trimethoprim-sulfamethoxazole, tigecycline, polymyxin, and meropenem/sulbactam are commonly used in the targeted treatment; (iv) the treatment effect was poor and affected by the treatment: timely active antimicrobial agents should be given. And untested antimicrobial agents represented by polymyxin and meropenem/sulbactam might be optional. Inactive antimicrobial agents must be replaced.

**KEYWORDS** carbapenem-resistant Gram-negative bacteria, healthcare-associated ventriculitis and meningitis, antimicrobial resistance, meropenem, polymyxin

Address correspondence to Guangzhi Shi, shiguangzhi@bjtth.org.

The authors declare no conflict of interest.

**G**ram-negative bacteria are common pathogens of health care-associated ventriculitis and meningitis (HCAVM) (1–3). Carbapenems, such as meropenem, are important antimicrobial agents used to treat Gram-negative bacteria-related infections (4, 5).

Carbapenem-resistant Gram-negative bacteria (CRGNB)-related HCAVM can be dangerous. Meropenem is recommended as the main empirical treatment for HCAVM, but resistance could lead to delayed treatment efficacy and adverse outcomes (6–8). Additionally, the proportion of CRGNB among all Gram-negative bacteria is increasing (9, 10). Thus, more patients with Gram-negative bacteria-related HCAVM are at risk. A delay in effective treatment leads to a longer duration of treatment and more antimicrobial agents, which increases the risk of adverse effects (11, 12).

To understand CRGNB-related HCAVM in neurosurgical patients, we aimed to report the antimicrobial resistance of the pathogens, as well as treatment and outcome. Moreover, how the treatment affected the ineffective treatment rate (ITR) was discussed.

## RESULTS

**Participants.** Overall, 92 cases of CRGNB-related HCAVM involving 91 patients were analyzed during the 9-year study period. For all cases, the mean age was $40.7 \pm 17.8$ years (range from 4 to 69 years), 44 (47.8%) were female, and 69 (75.0%) had solid tumors as the main diagnosis (Table 1).

**Bacterial spectrum.** There were 14 bacterial species identified in the 92 cases. *Acinetobacter baumannii* was the most common bacterial species, which was detected in 36 (39.1%) cases, followed by *Klebsiella pneumoniae*, which was detected in 31 (33.7%) cases (Fig. 1).

**Carbapenems resistance.** Of the 92 strains of CRGNB, 83 strains were tested against meropenem, and 68 (81.9%) were resistant. All 92 strains were tested against imipenem, and 90 (97.8%) were resistant (Table 2).

Among the 92 strains of CRGNB, 66 strains were resistant to meropenem and imipenem; 15 strains were susceptible to meropenem and resistant to imipenem; nine strains were not tested against meropenem but were resistant to imipenem; two strains were resistant to meropenem and susceptible to imipenem.

**Other antimicrobial agents resistance.** In addition to meropenem and imipenem, 24 antimicrobial agents were included in the antimicrobial susceptibility tests. The resistance rates were different for different antimicrobial agents. For most of the antimicrobial agents, the resistance rate was higher than 70.0%. Over 80% of CRGNB were tested against eight antimicrobial agents. The resistance rate range was 37.3%–85.9%; the trimethoprim-sulfamethoxazole resistance rate was the lowest, and the ceftazidime resistance rate was the highest. When all of the antimicrobial agents were considered, the resistance rate range was 11.6%–100.0%; the polymyxin resistance rate was the lowest, and the cefoxitin resistance rate was the highest. Moreover, the chloramphenicol, amikacin, levofloxacin, and tetracycline resistance rates were relatively low, ranging from 21.1% to 64.1% (Table 2). Eight strains, including three strains of *A. baumannii*, four strains of *K. pneumoniae*, and one strain of *Pseudomonas aeruginosa*, were resistant to all the antimicrobial agents tested. However, polymyxin, chloramphenicol, and tetracycline were not tested against these bacterial strains.

**Trends of antimicrobial resistance.** The resistance rates over the 9-year study period are shown (Appendix 1). There was no significant trend for all antimicrobial agents. Notably, in 2012–2014,2015–2017, and 2018–2020, the amikacin resistance rates were 61.9% (13/21), 56.0% (14/25), and 45.5% (15/33), respectively; the levofloxacin resistance rates were 66.7% (14/21), 80.0% (20/25), and 54.3% (25/46), respectively; and the trimethoprim-sulfamethoxazole resistance rates were 23.5% (4/17), 56.5% (13/23), and 32.6% (14/43), respectively (Table 2).

**Treatment.** In the initial empirical treatment stage, meropenem was used in 77 (83.7%) cases, and other antimicrobial agents were occasionally used. Moreover, $\beta$-lactamase inhibitors were used in some cases. Fourteen cases received adjusted empirical treatments (Appendix 2).

In the initial targeted treatment stage, meropenem was used in 50 (54.3%) cases. Trimethoprim-sulfamethoxazole, tigecycline, cefoperazone, etimicin, and levofloxacin

**TABLE 1** Clinical characteristics of cases in different groups

| Variables | Total (n = 92) | Group a (n = 41) | Group B (n = 12) | Group C (n = 39) | P value |
|---|---|---|---|---|---|
| Age (y), mean ±SD | 40.7 ± 17.8 | 40.2 ± 17.4 | 45.5 ± 17.9 | 39.7 ± 18.5 | 0.714 |
| Female, n (%) | 44 (47.8) | 22 (53.7) | 4 (33.3) | 18 (46.2) | 0.447 |
| Main diagnosis, n (%) | | | | | |
| Solid tumor | 69 (75.0) | 32 (78.0) | 7 (58.3) | 30 (76.9) | 0.357 |
| Vascular malformation | 10 (10.9) | 3 (7.3) | 1 (8.3) | 6 (15.4) | 0.488 |
| Traumatic brain injury | 10 (10.9) | 4 (9.8) | 3 (25.0) | 3 (7.7) | 0.231 |
| Other diseases[a] | 3 (3.3) | 2 (4.9) | 1 (8.3) | | 0.268 |
| Admission GCS[b], n (%) | | | | | |
| 13-15 | 85 (92.4) | 40 (97.6) | 10 (83.3) | 35 (89.7) | 0.188 |
| 9-12 | 2 (2.2) | | | 2 (5.1) | 0.249 |
| 3-8 | 5 (5.4) | 1 (2.4) | 2 (16.7) | 2 (5.1) | 0.160 |
| Surgery, n (%) | | | | | |
| Craniotomy | 81 (88.0) | 34 (82.9) | 10 (83.3) | 37 (94.9) | 0.223 |
| Transsphenoidal surgery | 10 (10.9) | 6 (14.6) | 2 (16.7) | 2 (5.1) | 0.310 |
| Repair | 1 (1.1) | 1 (2.4) | | | 0.533 |
| Chronic diseases[c], n (%) | 24 (26.1) | 9 (22.0) | 4 (33.3) | 11 (28.2) | 0.677 |
| Other bacteria[d], n (%) | 21 (22.8) | 8 (19.5) | 3 (25.0) | 10 (25.6) | 0.793 |
| Severe infection[e], n (%) | 6 (4.3) | 2 (4.9) | | 4 (10.3) | 0.385 |
| CSF[f] leak, n (%) | 25 (27.2) | 11 (26.8) | 3 (25.0) | 11 (28.2) | 0.974 |
| Incision infecion, n (%) | 14 (15.2) | 5 (12.2) | 2 (16.7) | 7 (17.9) | 0.765 |
| Bacteria species, n (%) | | | | | |
| *Acinetobacter baumannii* | 36 (39.1) | 9 (22.0) | 4 (33.3) | 23 (59.0) | <0.001 |
| *Klebsiella pneumoniae* | 31 (33.7) | 14 (34.1) | 5 (41.7) | 12 (30.8) | 0.781 |
| Other bacteria[g] | 25 (27.2) | 18 (43.9) | 3 (25.0) | 4 (10.3) | <0.001 |
| Ineffective treatment, n (%) | 42 (45.7) | 12 (29.3) | 12 (100.0) | 18 (46.2) | <0.001 |
| Poor outcome, n (%) | 51 (55.4) | 16 (39.0) | 12 (100.0) | 23 (59.0) | <0.001 |

[a]Including cerebral infarction, hemorrhage, and epilepsy.
[b]Glasgow Coma Scale.
[c]Hypertention in 18 cases, diabetes mellitus in three cases, and coronary heart disease in three cases, hyperthyroidism, hypothyroidism, rheumatoid arthritis, metabolic arthritis, and Hepatitis B. Five cases had two chronic diseases.
[d]Other bacteria isolated from cerebrospinal fluid cultures.
[e]Three abscesses and three ventriculitis.
[f]Cerebrospinal fluid.
[g]Including *Pseudomonas aeruginosa* (n = 9), *Enterobacter aerogenes* (n = 5), *Serratia marcescens* (n = 2), *Klebsiella oxytoca* (n = 1), *Escherichia coli* (n = 1), *Sphingobacterium multivorum* (n = 1), *Proteus rettgeri* (n = 1), *Morganella morganii* (n = 1), *Serratia plymuthica* (n = 1), *Sphingomonas paucimobilis* (n = 1), *Chryseobacterium indologenes* (n = 1), *Pseudomonas fluorescens* (n = 1).

were commonly used. Moreover, $\beta$-lactamase were are commonly used. Thirty cases received adjusted targeted treatments, and polymyxin was commonly used (Appendix 2).

**Outcome.** Overall, 42 (45.7%) cases had ineffective treatments, and 51 (55.4%) cases had poor outcomes (Table 1).

**Effect of treatment on the ITR.** All of the cases were divided into three groups according to treatment (Fig. 2). Table 1 shows the characteristics of the cases that belonged to the different groups, and Appendix 2 describes the treatments.

Forty-one cases received active antimicrobial agents (Group A), and the ITR was 29.3% (12/41); 39 cases received untested antimicrobial agents (Group C), and the ITR was 46.2% (18/39), and 12 cases received inactive antimicrobial agents (Group B). The ITR was 100.0% (12/12) ($P < 0.001$).

Of the cases that received active antimicrobial agents (Group A), 16 received them as initial empirical treatment, and the ITR was 18.8% (3/16); one received them as adjusted empirical treatment, and the ITR was 0 (0/1); 14 received them as initial targeted treatment, and the ITR was 28.6% (4/14); and 10 received them as adjusted targeted treatment, and the ITR was 50.0% (5/10) ($P = 0.172$).

Of the cases that received active antimicrobial agents (Group A), 17 received them starting on the first day of diagnosis, and the ITR was 23.5% (4/17); 12 received them starting at 2–10 days, and the ITR was 41.7% (5/12), and 12 received them starting at 11–57 days, and the ITR was 25.0% (3/12) ($P = 0.531$).

**Meropenem/sulbactam.** Overall, 13 cases received meropenem/sulbactam, including one in Group A and 12 in Group C. The ITR among cases who received meropenem/sulbactam

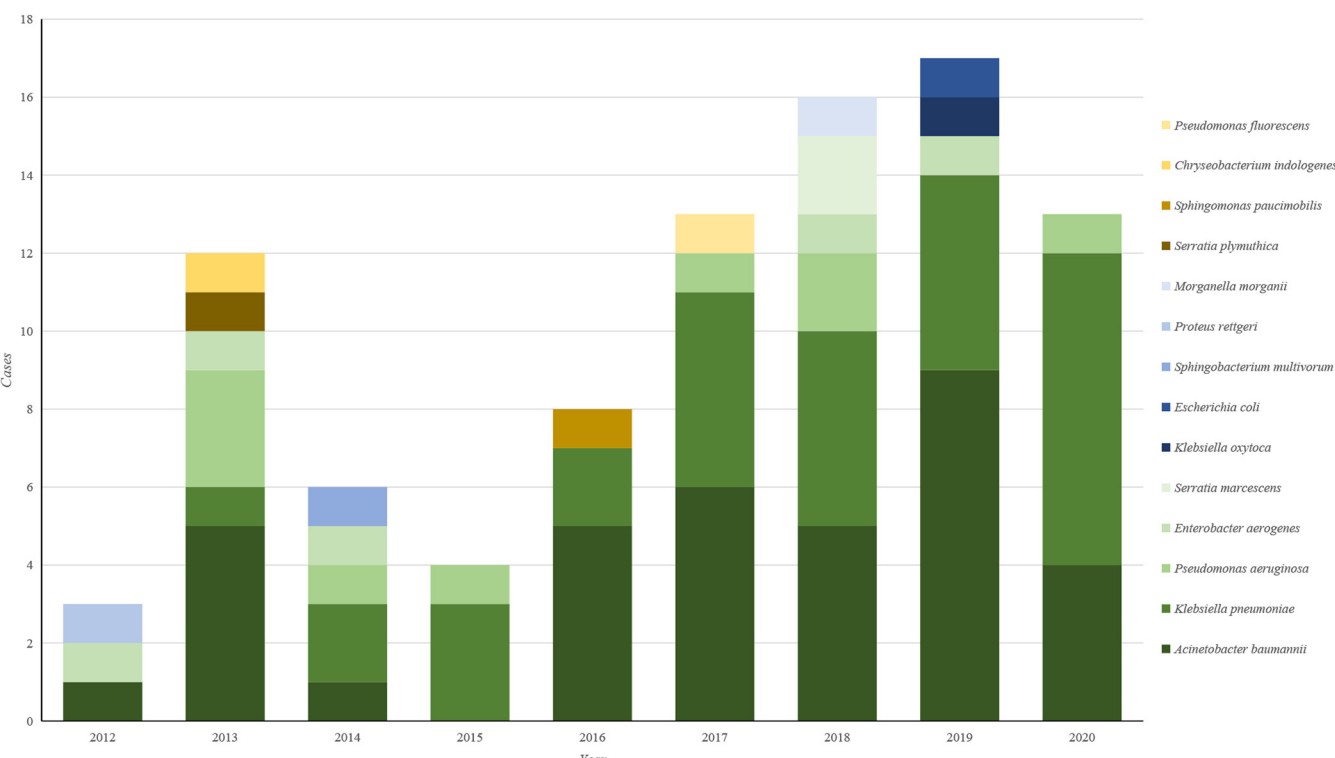

**FIG 1** The carbapenem-resistant Gram-negative bacteria causing health care-associated ventriculitis and meningitis in 2012–2020.

was 30.8% (4/13), and that among cases who did not receive meropenem/sulbactam was 48.1% (38/79) (*P* = 0.389).

Of the cases that received untested antimicrobial agents (Group C), 22 did not receive adjusted targeted treatment; insides, seven cases received meropenem/sulbactam as the initial targeted treatment, and the ITR was 28.5% (2/7); 13 cases received other antimicrobial agents as the initial targeted treatment, and the ITR was 46.2% (6/13) (*P* = 0.774).

**Polymyxin.** Overall, 19 cases received polymyxin, including seven in Group A and 12 in Group C. The ITR of cases who received polymyxin was 52.6% (10/19), and that of cases who did not receive polymyxin was 43.8% (32/73) (*P* = 0.607).

Of the cases in which untested antimicrobial agents were used (Group C), 17 received adjusted empirical treatment; insides, eight received polymyxin, the ITR was 37.5% (3/8); nine did not receive polymyxin, and the ITR was 55.6% (5/9) (*P* = 0.637).

**Local administration.** Overall, 13 cases received local administration (intrathecal or intraventricular administration), including one received meropenem, three received polymyxin, one received tigecycline, and one received polymyxin and tigecycline in Group A; four received polymyxin, two received tigecycline, and one received meropenem in Group C. The ITR was 46.1 (6/13). Seventy-nine cases did not receive local administration, and the ITR was 45.5% (36/79) (*P* = 1.000).

Of the 10 cases that received active antimicrobial agents as adjusted targeted treatment, four received local administration, and the ITR was 72.5% (3/4); six did not receive local administration, and the ITR was 33.3% (2/6) (*P* = 0.519). Of the cases that received untested antimicrobial agents (Group C), 17 received antimicrobial agents as adjusted targeted treatment, five received local administration, and the ITR was 40.0% (2/5); 12 did not receive local administration, and the ITR was 50.0% (6/12) (*P* = 1.000).

## DISCUSSION

Antimicrobial resistance was prevalent, and the resistance patterns were different for different antimicrobial agents. First, almost all CRGNB were resistant to imipenem,

**TABLE 2** The antimicrobial agents used in antimicrobial susceptibility tests and n (%) of resistant isolators in 2012–2014, 2015–2017, and 2018–2020

| Agents | Total | 2012–2014 | 2015–2017 | 2018–2020 | p-value[a] |
|---|---|---|---|---|---|
| Bacterial strains | 92 | 21 | 25 | 46 | |
| Meropenem | 83 | 21 | 24 | 38 | |
| Resistant, n (%) | 68 (81.9) | 15 (71.4) | 22 (91.7) | 31 (81.6) | 0.472 |
| Imipenem | 92 | 21 | 25 | 46 | |
| Resistant, n (%) | 90 (97.8) | 20 (95.2) | 24 (96.0) | 46 (100.0) | 0.175 |
| Amikacin | 79 | 21 | 25 | 33 | |
| Resistant, n (%) | 42 (53.2) | 13 (61.9) | 14 (56.0) | 15 (45.5) | 0.227 |
| Levofloxacin | 92 | 21 | 25 | 46 | |
| Resistant, n (%) | 59 (64.1) | 14 (66.7) | 20 (80.0) | 25 (54.3) | 0.179 |
| Polymyxin | 43 | 21 | 21 | 1 | |
| Resistant, n (%) | 5 (11.6) | 4 (19.0) | 1 (4.8) | 0 (0.0) | 0.147 |
| Trimethoprim-sulfamethoxazole | 83 | 17 | 23 | 43 | |
| Resistant, n (%) | 31 (37.3) | 4 (23.5) | 13 (56.5) | 14 (32.6) | 0.934 |
| Tetracycline | 35 | 16 | 18 | 1 | |
| Resistant, n (%) | 22 (62.9) | 9 (56.3) | 12 (66.7) | 1 (100.0) | 0.370 |
| Chloramphenicol | 19 | 10 | 8 | 1 | |
| Resistant, n (%) | 4 (21.1) | 3 (30.0) | 0 (0.0) | 1 (100.0) | 0.923 |
| Amoxicillin-clavulanate | 26 | 8 | 7 | 11 | |
| Resistant, n (%) | 25 (96.2) | 7 (87.5) | 7 (100.0) | 11 (100.0) | 0.188 |
| Ampicillin-sulbactam | 69 | 15 | 21 | 33 | |
| Resistant, n (%) | 55 (79.7) | 13 (86.7) | 17 (81.0) | 25 (75.8) | 0.378 |
| Ciprofloxacin | 86 | 21 | 25 | 40 | |
| Resistant, n (%) | 69 (80.2) | 15 (71.4) | 22 (88.0) | 32 (80.0) | 0.561 |
| Aztreonam | 51 | 14 | 14 | 23 | |
| Resistant, n (%) | 39 (76.5) | 10 (71.4) | 12 (85.7) | 17 (73.9) | 0.758 |
| Piperacillin | 67 | 21 | 24 | 22 | |
| Resistant, n (%) | 51 (76.1) | 15 (71.4) | 21 (87.5) | 15 (68.2) | 0.787 |
| Piperacillin-tazobactam | 87 | 21 | 24 | 42 | |
| Resistant, n (%) | 67 (77.0) | 14 (66.7) | 22 (91.7) | 31 (73.8) | 0.797 |
| Gentamicin | 78 | 21 | 25 | 32 | |
| Resistant, n (%) | 56 (71.8) | 15 (71.4) | 16 (64.0) | 25 (78.1) | 0.517 |
| Cefepime | 92 | 21 | 25 | 46 | |
| Resistant, n (%) | 76 (82.6) | 15 (71.4) | 23 (92.0) | 38 (82.6) | 0.427 |
| Cefotaxime | 37 | 17 | 19 | 1 | |
| Resistant, n (%) | 34 (91.9) | 14 (82.4) | 19 (100.0) | 1 (100.0) | 0.065 |
| Ceftazidime | 92 | 21 | 25 | 46 | |
| Resistant, n (%) | 79 (85.9) | 17 (81.0) | 23 (92.0) | 39 (84.8) | 0.845 |
| Ampicillin | 18 | 8 | 10 | 0 | |
| Resistant, n (%) | 17 (94.4) | 7 (87.5) | 10 (100.0) | | 0.264 |
| Ceftizoxime | 38 | 10 | 12 | 16 | |
| Resistant, n (%) | 35 (92.1) | 8 (80.0) | 11 (91.7) | 16 (100.0) | 0.071 |
| Cefoxitin | 10 | 0 | 0 | 10 | |
| Resistant, n (%) | 10 (100.0) | | | 10 (100.0) | |
| Tobramycin | 49 | 2 | 5 | 42 | |
| Resistant, n (%) | 36 (73.5) | 2 (100.0) | 2 (40.0) | 32 (76.2) | 0.684 |
| Nitrofurantoin | 43 | 2 | 5 | 36 | |
| Resistant, n (%) | 39 (90.7) | 2 (100.0) | 4 (80.0) | 33 (91.7) | 0.868 |
| Ceftriaxone | 33 | 0 | 0 | 33 | |
| Resistant, n (%) | 31 (93.9) | 0 | 0 | 31 (93.9) | |
| Cefuroxime | 33 | 0 | 0 | 33 | |
| Resistant, n (%) | 32 (97.0) | 0 | 0 | 32 (97.0) | |
| Cefotetan | 24 | 0 | 0 | 24 | |
| Resistant, n (%) | 22 (91.7) | 0 | 0 | 22 (91.7) | |

[a]For the trend of resistance rate.

while 18.1% of CRGNB were susceptible to meropenem. It is worth noting that *A. baumannii* and *K. pneumoniae*, the two most common CRGNB causing HCAVM, were rarely susceptible to meropenem. Of the 15 strains of meropenem-susceptible CRGNB in Group A, only one (6.7%) was *K. pneumoniae*.

The polymyxin resistance rate and chloramphenicol resistance rate were low; thus, these two antimicrobial agents could be used as empirical treatments in patients with

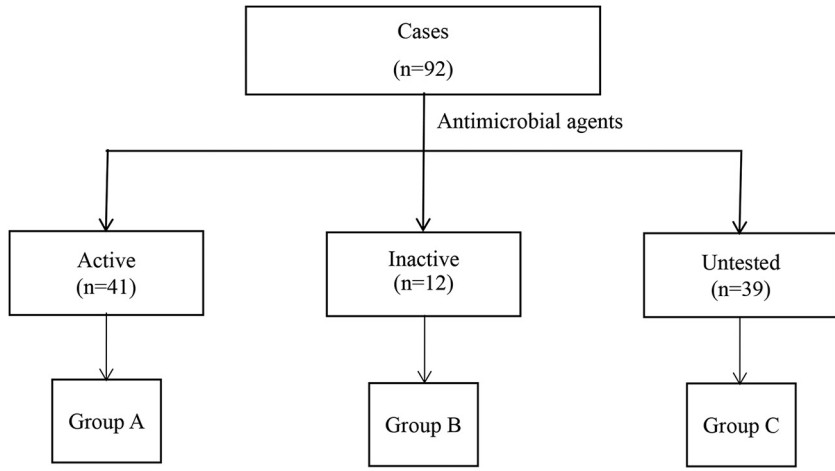

**FIG 2** Flow chart of groups selection. Antimicrobial agent: the antimicrobial agents with the highest priority used in the treatment based on antimicrobial susceptibility tests.

CRGNB-related HCAVM. Moreover, amikacin, levofloxacin, tetracycline, and trimethoprim-sulfamethoxazole are secondary antimicrobial agents that could be used in targeted treatment because of their relatively low resistance rates. Although eight strains of pan-resistant CRGNB were detected, polymyxin and chloramphenicol still could be used because the strains were not tested against these agents. Our finding was similar to that of a study that reported that polymyxin could be used for the treatment of CRGNB-related infection (13). However, tetracycline, another suitable treatment mentioned (13), should not be used as an empirical treatment for CRGNB-related HCAVM because of the high resistance rate.

The trends of antimicrobial resistance were different for different antimicrobial agents. We focused on antimicrobial agents for which bacteria have relatively low resistance rates. The amikacin resistance rate tended to decrease, while the resistance rates of levofloxacin and trimethoprim-sulfamethoxazole were unstable. Considering that the sample size was small and some trends were unstable, continuous and long-term monitoring is necessary. Fortunately, considering that the proportion of CRGNB among all Gram-negative bacteria is increasing (9, 10), the antimicrobial resistance pattern of CRGNB did not show an increasing trend.

Meropenem was the main empirical and targeted treatment, which is consistent with the recommendation of guideline (4). Other antimicrobial agents recommended in the guideline (4), including cefepime and ceftazidime, are rarely used as empirical treatments in our institute because of their high resistance rates. In the targeted treatment stage, active antimicrobial agents were most frequently used, followed by untested antimicrobial agents.

The outcomes of cases of CRGNB-related HCAVM were unacceptable. A total of 45.7% of cases had ineffective treatments, and 55.4% had poor outcomes. This conclusion is similar to those of previous studies (14, 15).

In the analysis of the effect of treatment on the outcome, we chose the ITR rather than the poor outcome rate as the study outcome because the sample sizes were too small to perform an adjusted analysis. The poor outcome rate is affected by many factors (16, 17); however, the ITR is mainly affected by the type of bacteria (4), the treatment (18), and the presence of immunodeficiency (19). The treatment was the exposure factor, and we did not include any patients with immune system diseases in the study. Therefore, adjusted analysis was not necessary.

Considering the priority ranking of the antimicrobial agents used for treatment, the ITR of the cases that received active antimicrobial agents was much lower than that of those that received untested or inactive antimicrobial agents. All cases that received only inactive antimicrobial agents for the whole course had ineffective treatment. Therefore, active antimicrobial agents should be used; untested antimicrobial agents in this study are optional, but inactive antimicrobial agents must be replaced.

Regarding the time to the administration of active antimicrobial agents, we found that the ITR was related to the relative and absolute time. The ITR was lower when the active antimicrobial agents were used earlier in relative time; however, for absolute time, although we did not find a similar relationship, the cases that received effective antimicrobial agents on the day of diagnosis had the lowest ITR. Therefore, active antimicrobial agents should be administered as soon as possible.

Meropenem/sulbactam is a choice for the treatment of CRGNB-related infection (20). Overall, the ITR of cases that received meropenem/sulbactam was lower than that of cases that did not receive this agent. Considering that meropenem/sulbactam was analyzed as an untested antimicrobial agent, we focused on the cases that received untested antimicrobial agents. Meropenem/sulbactam was mostly administered as an initial targeted treatment. After exclusion of the cases that received adjusted targeted treatment, we found that those who received meropenem/sulbactam as the initial targeted treatment had a decreased ITR. However, five of the 12 cases who received meropenem/sulbactam as the initial targeted treatment needed adjusted treatment. Therefore, meropenem/sulbactam could be used if an active antimicrobial agent is unavailable for initial targeted treatment, and continuous monitoring is necessary.

Polymyxin was the antimicrobial agent with the lowest resistance rate, and it is traditionally recommended for the treatment of CRGNB-related infections (13). Overall, the ITR of cases that received polymyxin was similar to that of cases that did not. However, only patients with adverse or poor effects received polymyxin; further analysis should be performed. In addition to its use as an active antimicrobial agent, polymyxin was widely used as an untested antimicrobial agent, especially in adjusted targeted treatments. The ITR of cases that received polymyxin as an untested antimicrobial agent as an adjusted targeted treatment was lower than that of cases that received other untested antimicrobial agents as adjusted treatments, as well as that of cases that received active antimicrobial agents as adjusted targeted treatments (37.5% [3/8] versus 50.0% [5/10], $P = 0.664$). Therefore, polymyxin is safe as an active or untested antimicrobial agent for treatment, especially when other antimicrobial agents are ineffective and adjusted targeted treatment is necessary.

Local administration is a treatment method in patients with CRGNB-related HCAVM (14, 18). However, the ITR of cases that received local administration was higher than that of cases that did not. In the subgroup analyses, we also did not find that local administration had a better treatment effect than that of intravenous administration. This conclusion is different from those of previous studies (14, 18). More randomized analyses are needed because the local administration could be the final method to salvage HCAVM.

There are several limitations to this study. First, this was a single-center retrospective study performed in a hospital, and further studies are needed. Second, not all antimicrobial agents, such as polymyxin, were tested for every strain of bacteria. The reason is that polymyxin was not a standing antimicrobial agent in our hospital, and the patients with the need for this treatment should outsource it. Third, the sample sizes were small, and subsequent adjusted analysis was difficult. Finally, the adverse effects and duration of treatment were not discussed.

## CONCLUSION

Antimicrobial resistance was prevalent but without increasing trends. Timely active antimicrobial agents are necessary. In addition, untested antimicrobial agents, represented by polymyxin and meropenem/sulbactam in this study, are optional. Inactive antimicrobial agents must be replaced.

## MATERIALS AND METHODS

**Study design.** This is a retrospective study performed in Beijing Tiantan Hospital, Capital Medical University (Beijing, China), a tertiary teaching hospital with one of the largest neurosurgical centers in China. All patients with CRGNB-related HCAVM in 2012–2020 were recruited. One case was one patient with one strain of bacteria, and the patient with two strains of bacteria was considered two cases. HCAVM was diagnosed according to the guideline (4).

**Cultures and antimicrobial susceptibility tests.** Cerebrospinal fluid (CSF) specimens were collected from patients with suspected HCAVM and incubated until flagged as positive or for 5 days in Bactec 9240 (Becton, Dickinson, America) in January 2012 -September 2018 or BacT/Alert 3D (bioMérieux, France) in October 2018 -December 2020. The positive CSF cultures were Gram-stained and sub-cultured onto solid medium using standard protocols. The antimicrobial agents were tested for activity against the bacteria using disk diffusion and broth microdilution methods according to guidelines from the Clinical and Laboratory Standards Institute. The techniques from the newest editions in the corresponding times were employed.

The interpretive categories were defined according to the Clinical and Laboratory Standards Institute M100, 31st edition guideline (21). The intermediate or susceptible-dose dependent strains were analyzed as susceptible strains since the antimicrobial agents were optional in clinical practice. The active antimicrobial agents were defined as agents to which the bacterial strains were susceptible or susceptible-dose dependent (or intermediate); the inactive antimicrobial agents were defined as agents to which the bacterial strains were resistant. Untested antimicrobial agents were defined as agents for which the antimicrobial susceptibility tests were not performed in some positive CSF cultures because the agents were not standing agents in our hospital or the bacteria isolated from CSF cultures should not be considered 'susceptible' to the agents according to the Clinical and Laboratory Standards Institute guidelines since the antimicrobial susceptibility tests were used to guide the treatment for HCAVM.

**Treatment.** Empirical treatment was defined as treatment initiated at diagnosis, and targeted treatment was defined as treatment administered after culture and antimicrobial susceptibility tests results were received. Adjusted treatment was defined as a change in the empirical or targeted treatment. Finally, treatment was divided into four stages: initial empirical treatment, adjusted empirical treatment, initial targeted treatment, and adjusted targeted treatment.

The antimicrobial agents used for treatment were classified, in order of priority, as active antimicrobial agents, untested antimicrobial agents, or inactive antimicrobial agents based on antimicrobial susceptibility tests independently during the four treatment stages. If the patient received an antimicrobial agent with a higher priority in a stage, antimicrobial agents with a lower priority were not discussed during that stage.

**Outcome.** The treatment effect was dichotomized into effective treatment and ineffective treatment according to guideline (4). Treatment was considered effective when HCAVM-related parameters gradually returned to the normal levels; HCAVM-related parameters included CSF parameters, CSF culture, and clinical parameters, in order of descending importance. Treatment was considered ineffective when HCAVM-related parameters, especially CSF parameters, did not return to normal levels.

The clinical outcome was dichotomized into poor outcome (Glasgow Outcome Scale 1–3) and acceptable outcome (Glasgow Outcome Scale 4–5) (22).

Treatment effect and clinical outcome were determined on the day of discharge from the hospital in which the patient with HCAVM was treated. Unplanned readmission within 1 month was considered a continuation of the previous hospitalization.

**Effect of treatment on the ITR.** We divided the cases into three groups according to the antimicrobial agents used for treatment based on antimicrobial susceptibility tests. The exposure factor was the treatment, and the study outcome was the ITR.

In addition to the data mentioned above, demographic characteristics, basic health information, surgical history, infection-related information, and intensive care unit admission data were collected.

**Statistical analysis.** Categorical variables are presented as frequencies and percentages. Continuous variables are described using means and standard deviations. Statistical analyses were performed using the R Programming Language version 4.0.2 and SPSS version 23. The characteristics of the groups were compared using contingency analysis or Fisher's exact test for categorical variables and the Kruskal–Wallis rank-sum test for continuous variables. The Chi-square test was used to detect trends in resistance rates. $P$ values $<0.05$ were significant.

**Ethics approval and consent to participate.** The study was retrospective and observational.

**Data availability.** Any data-related question should be directed to the corresponding author.

## SUPPLEMENTAL MATERIAL

Supplemental material is available online only.

**SUPPLEMENTAL FILE 1**, PDF file, 0.1 MB.

## ACKNOWLEDGMENTS

G.S. conceived the idea, G.S. and Y.Y. designed the work; Y.Y., Y.K., and J.M. implemented the data collection; Y.Y. did the statistical analysis; Y.Y. drafted the manuscript; and G.S. provided the critical revision.

This work was supported by the 'the Clinical Key Special Subject of Beijing Municipal Health Commission' (2100199). The funder had no role in study design, data analysis, the preparation or approval of the manuscript, or the decision to submit the manuscript for publication.

We have no competing interests to declare.

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
