## [Reviewer comments · Microbiology Spectrum]

Microbiology Spectrum

Carbapenem-resistant gram-negative bacteria-related healthcare-associated ventriculitis and meningitis: antimicrobial resistance of the pathogens, treatment, and outcome

Yi Ye, Yueyue Kong, Jiawei Ma, and Guangzhi Shi

Corresponding Author(s): Guangzhi Shi, Beijing Tian Tan Hospital

Review Timeline:

Submission Date:	January 21, 2022
Editorial Decision:	February 21, 2022
Revision Received:	March 11, 2022
Editorial Decision:	March 22, 2022
Revision Received:	March 30, 2022
Accepted:	April 8, 2022

Editor: Tomefa Asempa

Reviewer(s): Disclosure of reviewer identity is with reference to reviewer comments included in decision letter(s). The following individuals involved in review of your submission have agreed to reveal their identity: Istemi Seri (Reviewer #2); Liang Gao (Reviewer #3)

Transaction Report:

DOI: <https://doi.org/10.1128/spectrum.00253-22>

February 21, 2022

Prof. Guangzhi Shi
Beijing Tian Tan Hospital
No.119, South Fourth Ring West Road, Fengtai District, Beijing
Beijing, Beijing 105000
China

Re: Spectrum00253-22 (Carbapenem-resistant gram-negative bacteria-related healthcare-associated ventriculitis and meningitis: antimicrobial resistance of the pathogens, treatment, and outcome)

Dear Prof. Guangzhi Shi:

Link Not Available

Sincerely,

Tomefa Asempa

Journals Department
Editor Comments:

1. I would strongly advise use of editing services to refine paper for broader audience. <https://journals.asm.org/language-editing-services>
2. You will have to use the "compare" function in Microsoft word to see the edits made by the reviewers in the 2 PDFs attached.

Reviewer comments:

Reviewer #1 (Comments for the Author):

This is an interesting study on meningitis and ventriculitis by carbapenem-resistant Gram-negative bacteria. The bacterial strains described will probably represent the future in many countries of the world, which at present have less problems with antimicrobial resistance. I have several suggestions to improve the manuscript:

1. The authors must explain the "ineffective treatment rate" upon first appearance in the Abstract and text, and they must provide a clear definition in the Methods section.
2. The language needs revision to make the manuscript more understandable. Here are several examples:

- line 25. "fatal" probably means "has a poor outcome". Not all cases are fatal.
 - "accepted", one of the most frequently used words of this manuscript, probably must be replaced by "received".
 - line 47. Bacteria are susceptible, not antibiotics.
 - line 89. Probably "was considered as" must be used instead of "had".
 - line 115/116. What do the authors mean by "If the patient accepted antimicrobial agents with higher priority, antimicrobial agents with lower priority were not discussed"?
 - line 125. Does "judgement" mean "GOS"?
 - line 139. "Fisher's" must be capitalized, because it is a name.
 - lines 268/269. What is meant by "The treatment was the exposure"?
 - line 309. "a" should be replaced by "one" or "a single".
3. I find it very problematic to opt for the use of "not-tested antimicrobial agents" as "optional" (Abstract, line 48, and Discussion, lines 274/275, lines 298/299). Of course, the chance to hit the bacterium is greater, when you use an antibiotic, which was not tested, instead of an antibiotic tested resistant, but the goal should be to test the susceptibility of all potentially effective antibiotics in these infections.
4. line 120. Please include the exact definitions of "effective" and "ineffective".
5. lines 171-174 and 239-241. It is a pity that the 8 strains "resistant to all antimicrobial agents tested" were not tested for polymyxin, chloramphenicol and tetracyclines. The authors should at least explain why!
6. 260/261. Please clarify the difference between "ineffective treatment" and "poor outcome".
7. Did the authors use intrathecal therapy? If yes, which compounds were administered into the CSF, and what were the authors' experiences?

Reviewer #2 (Comments for the Author):

Thank you for giving me the opportunity to review this well-designed study,
Before publishing, I would like to make a few suggestions;

- 1) The last sentence of the abstract is very assertive, I think it is necessary to complete the abstract with a sentence that will make this subject a discussion point for future literature,
- 2) It is necessary to add a reference about the Glasgow score,

Apart from that, I marked minor corrections in the text,

Best regards

Reviewer #3 (Comments for the Author):

1. A native speaker is highly recommended to improve the language. Several sentences should be modified properly for they bring obscure understanding.
2. The majority cases of this research were diagnosed with one strain, however, multi-strains infected were the most scenario in clinical, the interaction between Gram strains and the overlapped antibiotics resistance need to be considered when drawing such a conclusion.
3. It will be more convincing whether the route of administration, such as systemic intravenous, intraventricular medication or combination of both, was compared to evaluate.

Staff Comments:

Preparing Revision Guidelines

For complete guidelines on revision requirements, please see the journal Submission and Review Process requirements at

<https://journals.asm.org/journal/Spectrum/submission-review-process>. **Submissions of a paper that does not conform to Microbiology Spectrum guidelines will delay acceptance of your manuscript. "**

Please return the manuscript within 60 days; if you cannot complete the modification within this time period, please contact me. If you do not wish to modify the manuscript and prefer to submit it to another journal, please notify me of your decision immediately so that the manuscript may be formally withdrawn from consideration by Microbiology Spectrum.

Title: Carbapenem-resistant gram-negative bacteria-related healthcare-associated ventriculitis and meningitis: antimicrobial resistance of the pathogens, treatment, and outcome

Authors: M.M. Yi Ye¹, M.M. Yueyue Kong¹, M.M. Jiawei Ma¹, M.D. Guangzhi Shi¹

1: Department of Critical Care Medicine, Beijing Tiantan Hospital, Capital Medical University, Beijing, China

Correspondence to: M.D. Guangzhi Shi, Department of Critical Care Medicine, Beijing Tiantan Hospital, Capital Medical University, Beijing, China. Telephone and fax number: 010-59978585. Postal address: No.119, South Fourth Ring West Road, Fengtai District, Beijing. E-mail: shiguangzhi@bjtth.org

Running title: CRGNB-related HCAVM.

Abstract

Background: Carbapenem-resistant gram-negative bacteria (CRGNB)-related

healthcare-associated ventriculitis and meningitis (HCAVM) are fatal. We aimed to report the antimicrobial resistance of the pathogens, treatment and outcome.

Methods: All neurosurgical patients with CRGNB-related HCAVM in 2012-2020 were recruited. The antimicrobial agents were divided into susceptible, not-tested, and resistant based on antimicrobial susceptibility tests. The treatment was divided into empirical treatment and targeted treatment according to the report of cerebrospinal fluid culture and antimicrobial susceptibility tests.

Results: Overall, 92 cases were recruited. For most antimicrobial agents, the resistance rate was higher than 70.0%. Polymyxin was the only antimicrobial agent with a resistance rate lower than 20.0%. And the chloramphenicol, trimethoprim-sulfamethoxazole, amikacin, levofloxacin, and tetracycline resistance rates were relatively low, with a range of 21.1%-64.1%. The meropenem resistance rate was 81.9%. There was no significant trend for any antimicrobial agent tested. Meropenem was the commonest antimicrobial agent used in the empirical treatment, trimethoprim-sulfamethoxazole and polymyxin were commonly used as susceptible antimicrobial agents, and meropenem sulbactam and polymyxin were commonly used as not-tested antimicrobial agents in the targeted treatment. Finally, 42 (45.7%) cases had ineffective treatments. The ITR of cases accepted susceptible antimicrobial agents was lower than that of cases accepted not-tested antimicrobial agents and cases accepted resistant antimicrobial agents (29.3% [12/41] vs. 46.2% [18/39] vs. 100.0% [12/12], $p < 0.001$).

Conclusion: The antimicrobial resistance is severe without increasing trend. Susceptible antimicrobial agents are necessary. And not-tested antimicrobial agents represented by polymyxin and meropenem sulbactam are optional. Resistant antimicrobial agents must be changed.

Keywords: Carbapenem-resistant gram-negative bacteria; Healthcare-associated ventriculitis and meningitis; Antimicrobial resistance; Meropenem; Polymyxin.

Introduction

Gram-negative bacteria are the common pathogens in healthcare-associated ventriculitis and meningitis (HCAVM) [1-3]. Carbapenems represented by meropenem are important antimicrobial agents used in gram-negative bacteria-related infections [4, 5].

The carbapenem-resistant gram-negative bacteria (CRGNB)-related HCAVM could be fatal. First, meropenem is recommended as the main empirical treatment to HCAVM, and the resistance could lead to delaying effective treatment and adverse

outcomes [6-8]. Second, as reported, the portion of CRGNB in gram-negative bacteria is increasing [9, 10]. Thus, more patients with gram-negative bacteria-related HCAVM are threatened. Third, delaying effective treatment means a longer duration of treatment and more antimicrobial agents, followed by adverse effects [11, 12].

To understand the CRGNB-related HCAVM in neurosurgical patients, we aimed to report the antimicrobial resistance of the pathogens, treatment, and outcome. Moreover, how the treatment affected the ineffective treatment rate (ITR) was discussed.

Methods

Study design

This is a retrospective study performed in Beijing Tiantan Hospital, Capital Medical University (Beijing, China), a tertiary teaching hospital with one of the largest neurosurgical centers in China. All patients with CRGNB-related HCAVM in 2012-2020 were recruited. One case was one patient with one strain of bacteria, and the patient with two strains of bacteria had two cases. HCAVM was diagnosed according to the guideline [4].

Cultures and antimicrobial susceptibility tests

Cerebrospinal fluid (CSF) specimens were collected from patients with suspected HCAVM and incubated until flagged as positive or for five days in BACTEC 9240 (Becton Dickinson, America) in January 2012 - September 2018 or BacT/ALERT 3D (bioMérieux, France) in October 2018 - December 2020. The broth was analyzed automatically every 10 minutes bacteria from positive bottles, was Gram-stained, and sub-cultured onto solid medium using standard protocols. The antimicrobial agents were tested for activity against the bacteria using disk diffusion and broth micro-

dilution methods according to the Clinical and Laboratory Standards Institute. The techniques from the newest editions in the corresponding times were employed.

The interpretive categories were defined according to the Clinical and Laboratory Standards Institute M100, 31st edition [13]. All intermediate or susceptible-dose dependent strains were analyzed as susceptible strains since the antimicrobial agent was optional in clinical practice.

Treatment

The empirical treatment was defined as the treatment the patient accepted since the diagnosis, and the targeted treatment means the treatment the patient accepted since the report of culture and antimicrobial susceptibility tests. The adjusted treatment means the treatment changed in the empirical or targeted treatment. Finally, the treatment was divided into four stages: initial empirical treatment, adjusted empirical treatment, initial targeted treatment, and adjusted targeted treatment.

The antimicrobial agents used in the treatment had a priority level from susceptible antimicrobial agents, to not-tested antimicrobial agents, to resistant antimicrobial agents based on antimicrobial susceptibility tests independently in the four treatment stages. If the patient accepted antimicrobial agents with higher priority, antimicrobial agents with lower priority were not discussed. The same-type antimicrobial agents were meropenem and imipenem, amikacin and etimicin, tetracycline and tigecycline in this study.

Outcome

The treatment was dichotomized into effective and ineffective according to the guideline [4]. The judgment has a priority level from CSF parameters, to CSF cultures, to clinical parameters.

The clinical outcome was dichotomized into poor (Glasgow Outcome Scale 1-3)

and acceptable (Glasgow Outcome Scale 4-5).

The judgment was determined on the discharge day from the hospitalization in which the patient had HCAVM. Unplanned readmission within one month was recognized as a continuation of the previous hospitalization.

How treatment affected the ITR

We divided the cases into three groups according to antimicrobial agents used in treatment based on antimicrobial susceptibility tests. The exposure was the treatment, and the study outcome was the ITR.

Apart from the data mentioned above, the demographic characteristics, basic health information, surgical history, infection-related information, and intensive care unit admission were described.

Statistical analysis

The categorical variables are presented as frequencies and percentages. The continuous variables are described using means and standard deviations. Statistical analyses were performed using the R Programming Language version 4.0.2. The characteristics of groups were compared using the contingency analysis or fisher's exact test for categorical variables and the Kruskal-Wallis rank-sum test for continuous variables. The Cox-Stuart test was used to detect trends of resistance rates. P-values <0.05 were significant.

Results

Participants

Overall, 92 cases with CRGNB-related HCAVM involving 91 patients were recruited in the nine years. For all of the cases, the mean age was 40.7 ± 17.8 years (range from 4 to 69 years), 44 (47.8%) cases were female, and 69 (75.0%) had solid tumors as the main diagnosis (Table 1).

Bacteria spectrum

There were 14 bacterial species in the 92 cases. *Acinetobacter baumannii* was the commonest bacteria species, which existed in 36 (39.1%) cases, followed by *Klebsiella pneumoniae*, which existed in 31 (33.7%) cases (Figure 1).

Antimicrobial resistance of carbapenems

In the 92 strains of CRGNB, meropenem was tested in 83 strains, 68 (81.9%) were resistant; and imipenem was tested in 92 strains, 90 (97.8%) were resistant (Table 2).

In the 92 strains of CRGNB, 66 strains were resistant to meropenem and imipenem; 15 strains were susceptible to meropenem and resistant to imipenem; nine strains were meropenem not-test and resistant to imipenem, and two strains were resistant to meropenem and susceptible to imipenem.

Antimicrobial resistance of other antimicrobial agents

Apart from meropenem and imipenem, 24 antimicrobial agents were used in the antimicrobial susceptibility tests. The resistance rates were different for different antimicrobial agents. For most of the antimicrobial agents, the resistance rate was higher than 70.0%. Eight antimicrobial agents were tested in over 80% of CRGNB. The range of resistance rates was 37.3%-85.9%. The trimethoprim-sulfamethoxazole resistance rate was the lowest, and the ceftazidime resistance rate was the highest. If all of the antimicrobial agents were considered, the range of resistance rates was 11.6%-100.0%, the polymyxin resistance rate was the lowest, and the cefoxitin resistance rate was the highest. Moreover, the chloramphenicol, amikacin, levofloxacin, and tetracycline resistance rates were relatively low, with a range of 21.1%-64.1% (Table 2). Eight strains, including three strains of *A. baumannii*, four strains of *K. pneumoniae*, one strain of *Pseudomonas aeruginosa* were resistant to all antimicrobial agents tested in related bacteria. However, polymyxin, chloramphenicol,

and tetracycline were not tested against them.

Trends of antimicrobial resistance

The resistance rates in the nine years are shown (Appendix 1). The trends were different for different antimicrobial agents. Significantly, in 2012-2014, 2015-2017, and 2018-2020, the amikacin resistance rate was 61.9% (13/21), 56.0% (14/25), and 45.5% (15/33), respectively; the levofloxacin resistance rate was 66.7% (14/21), 80.0% (20/25), and 54.3% (25/46), respectively; and the trimethoprim-sulfamethoxazole resistance rate 23.5% (4/17), 56.5% (13/23), and 32.6 (14/43), respectively (Table 2).

Treatment

In the initial empirical treatment stage, meropenem was used in 77 (83.7%) cases, and other antimicrobial agents were occasionally used. Moreover, β -lactamase inhibitors were used in some cases. Fourteen cases accepted adjusted empirical treatments (Appendix 2).

In the initial targeted treatment stage, meropenem was used in 50 (54.3%) cases. Trimethoprim-sulfamethoxazole, tigecycline, cefoperazone, etimicin, and levofloxacin were relatively commonly used. Moreover, β -lactamase inhibitors were commonly used. Thirty cases accepted adjusted targeted treatments, and polymyxin was commonly used (Appendix 2).

Outcome

Overall, 42 (45.7%) cases had ineffective treatments, and 51 (55.4%) cases had poor outcomes (Table 1).

How treatment affected ITR

All of the cases were divided into three groups according to the treatment (Figure 2). Table 1 shows the characteristics of the cases that belonged to the different groups,

and Appendix 2 describes the treatment.

Forty-one cases accepted susceptible antimicrobial agents (Group A), the ITR was 29.3% (12/41); 39 cases accepted not-tested antimicrobial agents (Group C), the ITR was 46.2% (18/39); and 12 cases accepted resistant antimicrobial agents (Group B), the ITR was 100.0% (12/12) ($p < 0.001$).

For the cases accepted susceptible antimicrobial agents (Group A), 16 cases accepted since initial empirical treatment, the ITR was 18.8% (3/16); one case accepted since adjusted empirical treatment, the ITR was 0 (0/1); 14 cases accepted since initial targeted treatment, the ITR was 28.6% (4/14); and ten cases accepted since adjusted targeted treatment, the ITR was 50.0% (5/10) ($p = 0.172$).

For the cases accepted susceptible antimicrobial agents (Group A), 17 cases accepted in the first day from the diagnosis, the ITR was 23.5% (4/17); 12 cases accepted in 2-10 days, the ITR was 41.7% (5/12); and 12 cases accepted in 11-57 days, the ITR was 25.0% (3/12) ($p = 0.531$).

Meropenem sulbactam

Overall, 13 cases accepted meropenem sulbactam, including one case in Group A, 12 cases in Group C. The ITR of cases accepted meropenem sulbactam was 30.8% (4/13), and not accepted was 48.1% (38/79) ($p = 0.389$).

In the cases accepted not-tested antimicrobial agents (Group C), 22 cases did not accept adjusted targeted treatment; inside, seven cases accepted meropenem sulbactam as initial targeted treatment, and the ITR was 28.5% (2/7), 13 cases accepted other antimicrobial agents as initial targeted treatment, the ITR was 46.2% (6/13) ($p = 0.774$).

Polymyxin

Overall, 19 cases accepted polymyxin, including seven cases in Group A and 12

cases in Group C. The ITR of cases accepted polymyxin was 52.6% (10/19), and not accepted was 43.8% (32/73) ($p=0.607$).

In the cases accepted not-tested antimicrobial agents (Group C), 17 cases accepted adjusted empirical treatment, inside, eight accepted polymyxin, the ITR was 37.5% (3/8), nine did not accept polymyxin, and the ITR was 55.6% (5/9) ($p=0.637$).

Discussion

The antimicrobial resistance was severe and different for different antimicrobial agents. First, almost all CRGNB were resistant to imipenem, while 18.1% of CRGNB were susceptible to meropenem. It is worth noting that *A. baumannii* and *K. pneumonia*, the two commonest bacteria species in CRGNB causing HCAVM, were rarely susceptible to meropenem. In the 15 cases with meropenem-susceptible CRGNB (Group A), only one (6.7%) was with *K. pneumonia*. The polymyxin resistance rate and chloramphenicol resistance rate were low, and the two antimicrobial agents could be used in the empirical treatment if the patients had CRGNB-related HCAVM. Moreover, amikacin, levofloxacin, tetracycline, and trimethoprim-sulfamethoxazole are underlying antimicrobial agents used in the targeted treatment because of relatively low resistance rates. Although there were eight strains of pan-resistant CRGNB, polymyxin and chloramphenicol still could be used because they were not tested. Our finding was similar to the study that reported that polymyxin could treat CRGNB-related infection [14]. However, tetracycline, which was seen as another suitable treatment like polymyxin [14], should not be used as an empirical treatment for CRGNB-related HCAVM because of the high resistance rate.

The trends of antimicrobial resistance were different for different antimicrobial agents. We focused on antimicrobial agents with relatively low resistance rates. For

amikacin, the resistance rate tended to decrease. And for levofloxacin and trimethoprim-sulfamethoxazole, the trends of resistance rates were unstable. Considering the sample size was small and trends were unstable, continuous and long-term monitoring is necessary. Fortunately, in the background that the portion of CRGNB in gram-negative bacteria is increasing [9, 10], the antimicrobial resistance of CRGNB did not show an increasing trend.

Meropenem was the main empirical and targeted treatment, which is consistent with the recommendation of guideline [4]. Other antimicrobial agents recommended in the guideline [4], including cefepime and ceftazidime, were rarely used as empirical treatment in our institute because of the high resistance rates. In the targeted treatment stage, susceptible antimicrobial agents were used widely, followed by not-tested antimicrobial agents.

The outcome of patients with CRGNB related HCAVM was unacceptable. 45.7% of cases had ineffective treatment, and 55.4% had poor outcomes. The conclusion is similar to previous studies [15, 16].

In the analysis that how the treatment affected the outcome, we chose the ITR rather than the poor outcome rate as the study outcome because the sample sizes were too small to perform an adjusted analysis. The poor outcome rate was affected by many factors [17, 18]. On the other hand, the ITR is mainly affected by the type of bacteria [4], the treatment [19], and the presence of immunodeficiency [20]. The treatment was the exposure, and we did not find immune-compromised patients in the study. Therefore, adjusted analysis was not necessary.

Concerning the highest priority for the antimicrobial agents used in the treatment, the ITR of the cases accepted susceptible antimicrobial agents was much lower than that of cases accepted not-tested or resistant antimicrobial agents. All cases only

accepted resistant antimicrobial agents for the whole course had ineffective treatments. Therefore, susceptible antimicrobial agents should be used, and not-tested antimicrobial agents are optional, but the resistant antimicrobial agents must be changed.

Concerning the time to accept susceptible antimicrobial agents, we found that the ITR was related to the relative and absolute time: the ITR was lower when the susceptible antimicrobial agents were used earlier in relative time; in absolute time, although we did not find the similar relationship, the cases accepted in the first day from diagnosis had the lowest ITR. Therefore, susceptible antimicrobial agents should be given as soon as possible.

Meropenem sulbactam is a choice in the CRGNB-related infection [21]. Overall, the ITR of cases accepted meropenem sulbactam was lower than those that did not accept it. Considering that meropenem sulbactam was analyzed as a not-tested antimicrobial agent, we focused on the cases accepted not-tested antimicrobial agents, the use of meropenem sulbactam concentrated on the initial targeted treatment, after exclusion of the cases accepted adjusted targeted treatment, we found the meropenem sulbactam used in the initial targeted treatment could significantly decrease the ITR (46.2% [6/13] vs. 28.5% [2/7], $p=0.774$). However, five in the 12 cases accepted meropenem sulbactam in the initial targeted treatment still needed to accept the adjusted treatment. Therefore, meropenem sulbactam could be used if the susceptible antimicrobial agent is unavailable in the initial targeted treatment, and continuous monitoring is necessary.

Polymyxin was the antimicrobial agent with the lowest resistance rate and is traditionally recommended for treating CRGNB-related infection[14]. Overall, the ITR of cases that accepted polymyxin was similar to that of cases did not accept it.

However, only patients with adverse or poor effects accepted polymyxin, therefore, more analysis should be performed. Apart from being used as a susceptible antimicrobial agent, polymyxin was widely used as a not-tested antimicrobial agent, especially in the adjusted targeted treatment. The ITR of cases accepted polymyxin as a not-tested antimicrobial agent since the adjusted targeted treatment was lower than that of not only the cases accepted other not-tested antimicrobial agents in the adjusted treatment, but cases accepted susceptible antimicrobial agents since the adjusted targeted treatment (37.5% [3/8] vs. 50.0% [5/10], $p=0.664$). Therefore, polymyxin is safe as a susceptible antimicrobial agent or a not-tested antimicrobial agent in treatment, especially when other antimicrobial agents are not helpful and adjusted targeted treatment is necessary.

There are several limitations to this study. First, this is a two-center retrospective study performed in a hospital, and further studies are needed. Second, not all antimicrobial agents were tested in every strain of bacteria, such as polymyxin. Third, the sample sizes were small, and following adjusted analysis was difficult. Finally, the adverse effects and the duration of treatment and were not discussed.

Conclusion

The antimicrobial resistance is severe without increasing trends. Timely susceptible antimicrobial agents are necessary. And not-tested antimicrobial agents represented by polymyxin and meropenem sulbactam are optional. Resistant antimicrobial agents must be changed.

Abbreviations

HCAVM: Healthcare-associated ventriculitis and meningitis; CRGNB: Carbapenem-resistant gram-negative bacteria; ITR: Ineffective treatment rate; CSF: Cerebrospinal fluid.

Acknowledgments

None.

Authors' contributions

GS conceived the idea, GS and YY designed the work, YY, YK, and JM implemented the data collection, YY did the statistical analysis, YY drafted the manuscript, GS provided the critical revision.

Funding

This work was supported by the 'Yangfan Plan of Beijing Municipal Hospital Administration' (ZYLX202109). The funder had no role in study design, data analysis, the preparation or approval of the manuscript, or the decision to submit the manuscript for publication.

Availability of data and materials

Any data-related question should be directed to the corresponding author.

Declarations

Ethics approval and consent to participate

The study was retrospective and observational.

Consent for publication

None.

Competing interests

All authors declared no competing interests.

References

- [1] Kourbeti IS, Vakis AF, Ziakas P, et al. Infections in patients undergoing craniotomy: risk factors associated with post-craniotomy meningitis. *J Neurosurg.* 122(5). United States,2015. 1113-9.
- [2] Srihawan C, Castelblanco RL, Salazar L, et al. Clinical Characteristics and

Predictors of Adverse Outcome in Adult and Pediatric Patients With Healthcare-Associated Ventriculitis and Meningitis. *Open Forum Infect Dis.* 3(2) ,2016. ofw077.

[3] Rogers T, Sok K, Erickson T, et al. Impact of Antibiotic Therapy in the Microbiological Yield of Healthcare-Associated Ventriculitis and Meningitis. *Open Forum Infect Dis.* 6(3) ,2019. ofz050.

[4] Tunkel AR, Hasbun R, Bhimraj A, et al. 2017 Infectious Diseases Society of America's Clinical Practice Guidelines for Healthcare-Associated Ventriculitis and Meningitis. *Clin Infect Dis.* 64(6) ,2017. e34-e65.

[5] Tamma PD, Aitken SL, Bonomo RA, et al. Infectious Diseases Society of America Guidance on the Treatment of Extended-Spectrum β -lactamase Producing Enterobacterales (ESBL-E), Carbapenem-Resistant Enterobacterales (CRE), and *Pseudomonas aeruginosa* with Difficult-to-Treat Resistance (DTR-P. *aeruginosa*). *Clin Infect Dis.* 72(7). United States,2021. e169-e183.

[6] Kyo M, Ohshimo S, Kosaka T, et al. Impact of inappropriate empiric antimicrobial therapy on mortality in pediatric patients with bloodstream infection: a retrospective observational study. *J Chemother.* 31(7-8). England,2019. 388-393.

[7] Tang Y, Wu X, Cheng Q, et al. Inappropriate initial antimicrobial therapy for hematological malignancies patients with Gram-negative bloodstream infections. *Infection.* 48(1). Germany,2020. 109-116.

[8] Cain SE, Kohn J, Bookstaver PB, et al. Stratification of the impact of inappropriate empirical antimicrobial therapy for Gram-negative bloodstream infections by predicted prognosis. *Antimicrob agents Chemother.* 59(1) ,2015. 245-50.

[9] Nordmann P, Poirel L. Epidemiology and Diagnostics of Carbapenem Resistance in Gram-negative Bacteria. *Clin Infect Dis.* 2019. 69(Suppl 7): S521-S528.

[10] Brink AJ. Epidemiology of carbapenem-resistant Gram-negative infections

globally. *Curr Opin Infect Dis.* 2019. 32(6): 609-616.

[11] Lee JD, Heintz BH, Mosher HJ, et al. Risk of acute kidney injury and *Clostridioides difficile* infection with piperacillin/tazobactam, cefepime and meropenem with or without vancomycin. *Clin Infect Dis.* United States,2020.

[12] Wardill HR, van der Aa S, da Silva Ferreira AR, et al. Antibiotic-induced disruption of the microbiome exacerbates chemotherapy-induced diarrhoea and can be mitigated with autologous faecal microbiota transplantation. *Eur J Cancer.* 153England,2021. 27-39.

[13] CLSI. Performance Standards for Antimicrobial Susceptibility Testing. 31st ed. CLSI supplement M100. Clinical and Laboratory Standards Institute; 2021.

[14] Doi Y. Treatment Options for Carbapenem-resistant Gram-negative Bacterial Infections. *Clin Infect Dis.* 2019. 69(Suppl 7): S565-S575.

[15] Chusri S, Sakarunchai I, Kositpantawong N, et al. Outcomes of adjunctive therapy with intrathecal or intraventricular administration of colistin for post-neurosurgical meningitis and ventriculitis due to carbapenem-resistant *Acinetobacter baumannii*. *Int J Antimicrob agents.* 2018. 51(4): 646-650.

[16] Rodríguez Guardado A, Blanco A, Asensi V, et al. Multidrug-resistant *Acinetobacter* meningitis in neurosurgical patients with intraventricular catheters: assessment of different treatments. *J Antimicrob Chemother.* 2008. 61(4): 908-13.

[17] Haniffa R, Mukaka M, Munasinghe SB, et al. Simplified prognostic model for critically ill patients in resource limited settings in South Asia. *Crit Care.* 21(1) ,2017. 250.

[18] Pirracchio R, Petersen ML, Carone M, et al. Mortality prediction in intensive care units with the Super ICU Learner Algorithm (SICULA): a population-based study. *Lancet Respir Med.* 3(1) ,2015. 42-52.

- [19] Pan S, Huang X, Wang Y, et al. (2018) Efficacy of intravenous plus intrathecal/intracerebral ventricle injection of polymyxin B for post-neurosurgical intracranial infections due to MDR/XDR *Acinetobacter baumannii*: a retrospective cohort study. *Antimicrob Resist Infect Control* 7:8
- [20] Jimenez A, Fennie K, Munoz-Price LS, et al. Duration of carbapenemase-producing Enterobacteriales carriage among ICU patients in Miami, FL: A retrospective cohort study. *Am J Infect Control*. 49(10). United States,2021. 1281-1286.
- [21] Mohd Sazly Lim S, Heffernan AJ, Zowawi HM, Roberts JA, Sime FB. Semi-mechanistic PK/PD modelling of meropenem and sulbactam combination against carbapenem-resistant strains of *Acinetobacter baumannii*. *Eur J Clin Microbiol Infect Dis*. 2021. 40(9): 1943-1952.

Figure legends

Figure 1. The carbapenem-resistant gram-negative bacteria causing healthcare-associated ventriculitis and meningitis in 2012-2020.

Figure 2. Flow chart of groups selection.

CRGNB: Carbapenem-resistant gram-negative bacteria;

HCAVM: Healthcare-associated ventriculitis and meningitis;

Agent: The agents used in the treatment based on antimicrobial susceptibility tests.

Response to Reviewers

Thanks for your professional comments. Before responding to your comments, we would like to mention the statistical method of the trends of the resistance rates. We changed the Cox-Stuart test to the Chi-square test. The two methods could be used to test trends, and we did not reach a significant result in either method. We chose the Chi-square because the method is more traditional, and the p-values same to be more diversified.

Editor Comments:

Comments: I would strongly advise use of editing services to refine paper for broader audience.

Response: *Thanks for your advice, and we have chosen aje as the language helper.*

Reviewer comments:

Reviewer #1

Comments: This is an interesting study on meningitis and ventriculitis by carbapenem-resistant Gram-negative bacteria. The bacterial strains described will probably represent the future in many countries of the world, which at present have less problems with antimicrobial resistance.

Response: *Thanks for your positive judgment, and we feel every suggestion is reasonable, which makes this manuscript more convincing,*

1. Comment: The authors must explain the "ineffective treatment rate"

upon first appearance in the Abstract and text, and they must provide a clear definition in the Methods section.

Response: *This is an important suggestion, helping the work convincing. The effective treatment means the HCAVM-related parameters gradually changed to the normal level. HCAVM-related parameters have a priority level from CSF parameters, CSF cultures, to clinical parameters. In the guideline of HCAVM, the response of treatment was judged based on clinical parameters, negative CSF cultures, and CSF parameters. The reason why we emphasize the CSF parameters is that the CSF parameters are the necessary indicators for improvement. For patients with external drainage, the CSF culture could still be positive when all the CSF parameters changed to the normal level. Finally, for neurosurgical adults, the main diagnosis could affect the clinical parameters such as headache. Therefore, we used the sequence to define the effective treatment.*

2. Comment: The language needs revision to make the manuscript more understandable. Here are several examples:

- line 25. "fatal" probably means "has a poor outcome". Not all cases are fatal.

Response: *Thanks for the advice; we feel "dangerous" could be a better expression.*

- "accepted", one of the most frequently used words of this manuscript, probably must be replaced by "received".

Response: *Thanks for the advice, "received" is a more appropriate word in this situation, and we have changed it.*

- line 47. Bacteria are susceptible, not antibiotics.

Response: *Thanks for the correction. After reanalyzing the expression, we concluded that the bacteria could be divided into susceptible and resistant, while the antimicrobial agent could be divided into sensitive and non-sensitive.*

- line 89. Probably "was considered as" must be used instead of "had".

Response: *Thanks for the comment. We have changed it.*

- line 115/116. What do the authors mean by "If the patient accepted antimicrobial agents with higher priority, antimicrobial agents with lower priority were not discussed"?

Response: *A patient could receive multi antimicrobial agents in a stage; the sensitive antimicrobial agent is the highest priority, followed by untested and non-sensitive antimicrobial agents. If a patient received the sensitive agent in a stage, the antimicrobial agents with lower priority were not discussed; if a patient received untested antimicrobial agents in a stage, the non-sensitive antimicrobial agent was not discussed.*

- line 125. Does "judgement" mean "GOS"?

Response: *GOS was a part of the judgment, and the treatment effect (effective or ineffective treatment) was judged simultaneously.*

- line 139. "Fisher`s" must be capitalized, because it is a name.

Response: *Thanks for your correction.*

- lines 268/269. What is meant by "The treatment was the exposure"?

Response: *The treatment was the exposure factor; we have changed the expression.*

- line 309. "a" should be replaced by "one" or "a single".

Response: *Thanks for the correction.*

3. Comment: I find it very problematic to opt for the use of "not-tested antimicrobial agents" as "optional" (Abstract, line 48, and Discussion, lines 274/275, lines 298/299). Of course, the chance to hit the bacterium is greater, when you use an antibiotic, which was not tested, instead of an antibiotic tested resistant, but the goal should be to test the susceptibility of all potentially effective antibiotics in these infections.

Response: *The suggestion is very important, and we do not want to make any misunderstanding. We emphasized two antimicrobial agents in this work; the polymyxin should be tested regularly. However, as an expensive antimicrobial agent, polymyxin is not a standing drug in our hospital. If the patients need the antimicrobial agent, they should outsource it. This is a realistic problem. Moreover, meropenem sulbactam is an experimental antimicrobial agent in our hospital as a part of a multi-center study. The antimicrobial susceptibility test has not been performed. We suggest that all potentially effective antibiotics should be tested, as you said, and if all the antimicrobial agents could not be considered, maybe the untested*

antimicrobial agents with the low resistance rates according to the antimicrobial susceptibility tests or the good treatment effects according to the clinical practice could be the last choice. In total, your suggestion must be taken highly to, and we will make all effort to benefit the patients.

4. Comment: line 120. Please include the exact definitions of "effective" and "ineffective".

Response: *The effective treatment means the HCAVM-related parameters gradually changed to the normal level, and all HCAVM-related parameters have a priority level from CSF parameters, CSF cultures, and clinical parameters. The ineffective treatment means the HCAVM-related parameters did not change to the normal level, especially the CSF parameters.*

5. Comment: lines 171-174 and 239-241. It is a pity that the 8 strains "resistant to all antimicrobial agents tested" were not tested for polymyxin, chloramphenicol and tetracyclines. The authors should at least explain why!

Response: *This is a limitation of this study. Polymyxin should be tested regularly. However, as an expensive antimicrobial agent, polymyxin is not a standing drug in our hospital. If the patients need the antimicrobial agent, they should outsource it. This is a realistic problem. Moreover, if the antimicrobial agents with low blood-CSF barrier transmittance, including tetracycline, should be tested when the bacteria are isolated*

from CSF cultures is still a focus of the debate. As for chloramphenicol, this is a less frequently used antimicrobial agent, which leads to the less frequently tested. According to the CLSI, when the bacteria isolated from CSF cultures, the antimicrobial agents administered by oral route only, 1st- and 2nd-generation cephalosporins and cephamycins, doripenem, ertapenem, imipenem, and lefamulin, clindamycin, macrolides, tetracyclines, and fluoroquinolones might be related to dangerously misleading results when they are tested and reported as susceptible. This expression might lead to the misunderstanding that the test against the antimicrobial agents is unnecessary. We are trying to change the view of colleagues in the laboratory because we could use more methods such as intraventricular injection.

6. Comment: 260/261. Please clarify the difference between "ineffective treatment" and "poor outcome".

Response: *The ineffective treatment was judged according to the treatment effect of HCAVM, and the clinical outcome was judged according to GOS. In this work, almost all cases with ineffective treatments had poor outcomes. And some cases with poor outcomes were caused by other situations.*

7. Comment: Did the authors use intrathecal therapy? If yes, which compounds were administered into the CSF, and what were the authors' experiences?

Response: *Thanks for your suggestion. In this study, we found 13 cases that received intrathecal or intraventricular administration. However, we did not find the advantages of the local administration; that is why we did not mention it in the first version of the manuscript. The reason could be the patients received the local administration faced more severe situations. More analysis is needed.*

Reviewer #2:

1. Comment: The last sentence of the abstract is very assertive, I think it is necessary to complete the abstract with a sentence that will make this subject a discussion point for future literature,

Response: *Thanks for the suggestion, and we did not want to lead to any misunderstanding. And we have changed the expression to “might be optional.”*

2. Comment: It is necessary to add a reference about the Glasgow score,

Response: *Thanks for the advice; the reference completed this manuscript.*

Reviewer #3:

1. Comment: A native speaker is highly recommended to improve the language. Several sentences should be modified properly for they bring obscure understanding.

Response: *Thanks for your advice, and we have chosen aje as the language helper.*

2. Comment: The majority cases of this research were diagnosed with one strain, however, multi-strains infected were the most scenario in clinical, the interact between Gram strains and the overlapped antibiotics resistance need to be considerate when draw such a conclusion.

Response: *Thanks for the suggestion; this situation did happen in this clinical practice; we mentioned we have 21 (22.8%) strains of bacteria that are faced with the situation with multi-strains infection. We reanalyzed the data and found every positive CSF culture was exclusive, which means we did not find that the two strains were cultured in a single positive CSF culture in this study. In this patient with two strains of the CRGNB in this work, the first positive cultures of the two strains are apart for eight days.*

3. Comment: It will be more convincing whether the route of administration, such as systemic intravenous, intraventricular medication or combination of both, was compared to evaluate.

Response: *Thanks for your suggestion. We found 13 cases that received intrathecal or intraventricular administration in this study. However, we did not find the advantages of the local administration; that is why we did not mention it in the first version of the manuscript. The reason could be the patients received the local administration faced more severe*

situations. More analysis is needed.

4. **Comment:** In discussion, paragraphs need further rearrangement to make each part discuss thoroughly.

Response: *Thanks for your suggestion. We have rearranged the discussion part, especially the resistance rate.*

March 22, 2022

Prof. Guangzhi Shi
Beijing Tian Tan Hospital
No.119, South Fourth Ring West Road, Fengtai District, Beijing
Beijing, Beijing 105000
China

Re: Spectrum00253-22R1 (Carbapenem-resistant gram-negative bacteria-related healthcare-associated ventriculitis and meningitis: antimicrobial resistance of the pathogens, treatment, and outcome)

Dear Prof. Guangzhi Shi:

Thank you for submitting your manuscript to Microbiology Spectrum. As you will see your paper is very close to acceptance. Please modify the manuscript along the lines I have recommended. As these revisions are quite minor, I expect that you should be able to turn in the revised paper in less than 30 days, if not sooner. If your manuscript was reviewed, you will find the reviewers' comments below.

When submitting the revised version of your paper, please provide (1) point-by-point responses to the issues I raised in your cover letter, and (2) a PDF file that indicates the changes from the original submission (by highlighting or underlining the changes) as file type "Marked Up Manuscript - For Review Only". Please use this link to submit your revised manuscript. Detailed instructions on submitting your revised paper are below.

Link Not Available

Sincerely,

Tomefa Asempa

Editor comments:

1. Antibiotics are not sensitive or non-sensitive. They can have activity or described as active or inactive. please revise paper accordingly.
2. meropenem sulbactam is a combination product so write as meropenem/sulbactam to avoid confusion.
3. Several grammatical errors. Please fix accordingly with this revision. No other opportunities will be given to fix and decision will be made accordingly.
4. list the untested antibiotics in the conclusion of the abstract.
5. provide definition of untested in the manuscript and why a drug would be untested.

Preparing Revision Guidelines

- point-by-point responses to the issues I raised in your cover letter
- Upload a compare copy of the manuscript (without figures) as a "Marked-Up Manuscript" file.
- Each figure must be uploaded as a separate file, and any multipanel figures must be assembled into one file.

- Manuscript: A .DOC version of the revised manuscript
- Figures: Editable, high-resolution, individual figure files are required at revision, TIFF or EPS files are preferred

Please return the manuscript within 60 days; if you cannot complete the modification within this time period, please contact me. If you do not wish to modify the manuscript and prefer to submit it to another journal, please notify me of your decision immediately so that the manuscript may be formally withdrawn from consideration by Microbiology Spectrum.

Response to Reviewers

Thanks for your professional comments. I felt the comments helped our work be persuasive, especially comment 1, which helped solve a serious problem.

Editor Comments:

Comment 1. Antibiotics are not sensitive or non-sensitive. They can have activity or described as active or inactive. please revise paper accordingly.

Response: It is a critical comment. Actually, we were puzzled about which words were suitable to describe it. Thanks a lot!

Comment 2. meropenem sulbactam is a combination product so write as meropenem/sulbactam to avoid confusion.

Response: We have changed it according to your comment.

Comment 3. Several grammatical errors. Please fix accordingly with this revision. No other opportunities will be given to fix and decision will be made accordingly.

Response: Thanks for your comment. We have changed them.

Comment 4. list the untested antibiotics in the conclusion of the abstract.

Response: We do not want to cause a misunderstanding. This comment is helpful.

Comment 5. provide definition of untested in the manuscript and why a drug would be untested.

Response: Thanks for your advice. In the Methods part, we have

described that Untested antimicrobial agents mean the antimicrobial susceptibility tests to the agents were not performed because the agents are not standing agents in our hospital, or the bacteria isolated from CSF cultures should not be considered 'susceptible' to the agents according to the Clinical and Laboratory Standards Institute guidelines since the antimicrobial susceptibility tests were used to guide the treatment for HCAVM.

April 8, 2022

Prof. Guangzhi Shi
Beijing Tian Tan Hospital
No.119, South Fourth Ring West Road, Fengtai District, Beijing
Beijing, Beijing 105000
China

Re: Spectrum00253-22R2 (Carbapenem-resistant gram-negative bacteria-related healthcare-associated ventriculitis and meningitis: antimicrobial resistance of the pathogens, treatment, and outcome)

Dear Prof. Guangzhi Shi:

Your manuscript has been accepted, and I am forwarding it to the ASM Journals Department for publication. You will be notified when your proofs are ready to be viewed.

Sincerely,

Tomefa Asempa
Editor, Microbiology Spectrum
